# Minimally Invasive Surgery for Cervical Cancer in Light of the LACC Trial: What Have We Learned?

Omar Touhami [1,]* and Marie Plante [2]

1   Gynecologic Oncology Division, Department of Obstetrics and Gynecology, Centre Intégré Universitaire de Santé et Services Sociaux CIUSSS du Saguenay—Lac-Saint-Jean, Sherbrooke University, Sherbrooke, QC J1K 2R1, Canada
2   Gynecologic Oncology Division, L'Hotel-Dieu de Québec, CHU de Québec, Laval University, Quebec, QC G1V 0A6, Canada; marie.plante.med@ssss.gouv.qc.ca
*   Correspondence: omar.touhami.med@ssss.gouv.qc.ca

**Abstract:** Cervical cancer is the most common gynecologic malignancy and the fourth most common cancer in women worldwide. Over the last two decades, minimally invasive surgery (MIS) emerged as the mainstay in the surgical management of cervical cancer, bringing advantages such as lower operative morbidity and shorter hospital stay compared to open surgery while maintaining comparable oncologic outcomes in numerous retrospective studies. However, in 2018, a prospective phase III randomized controlled trial, "Laparoscopic Approach to Carcinoma of the Cervix (LACC)", unexpectedly reported that MIS was associated with a statistically significant poorer overall survival and disease-free survival compared to open surgery in patients with early-stage cervical cancer. Various hypotheses have been raised by the authors to try to explain these results, but the LACC trial was not powered to answer those questions. In this study, through an exhaustive literature review, we wish to explore some of the potential causes that may explain the poorer oncologic outcomes associated with MIS, including the type of MIS surgery, the size of the lesion, the impact of $CO_2$ pneumoperitoneum, prior conization, the use of uterine manipulator, the use of protective measures, and the effect of surgical expertise/learning curve.

**Keywords:** cervical cancer; minimally invasive surgery; radical hysterectomy; robotic surgery; LACC trial; recurrences; conization; uterine manipulator; vaginal closure; colpotomy

## 1. Introduction

In the era of evidence-based medicine, healthcare professionals are overwhelmed with a plethora of studies and articles. In terms of hierarchy, randomized controlled trials (RCT) are at the top of the pyramid as they are thought to provide the highest level of evidence [1].

Indeed, it is considered that RCT offers the most reliable evidence on the effectiveness of interventions. In fact, the processes used during the design and conduct of these studies minimize the risk of bias and confounding factors, which in turn can influence the results [2]. Because of this, the findings produced by RCTs are likely to be closer to the real effect than the findings generated by other research methods [3].

It is then not surprising that the results of the LACC trial showing the inferiority of minimally invasive surgery (MIS) in the management of early-stage cervical cancer in terms of disease-free survival (DFS) and overall survival (OS) compared to laparotomy had a profound impact on our vision and perspective and dramatically changed our practice [4]. Indeed, in light of the findings of the LACC trial, ESGO and NCCN changed their recommendations in favor of laparotomy in the surgical management of cervical cancer [5,6].

However, besides the well-known limitations of RCT such as the possible incorrect statistical inference, the low internal or external validity, the misinterpretation of the difference in outcomes, and the publication bias [7], randomized controlled surgical trials

are essentially different from other types of RCT because of the variation in skills and surgical proficiency of participating surgeons and centers and also because it is difficult to blind a surgical procedure [8]. Additional challenges in the design, conduct, and analysis of randomized controlled surgical trials include ethical dilemmas, patients' preferences, variability in surgical proficiency and surgical technique, and finally, pre/postoperative care [9]. Indeed, criticisms regarding the methodology and possible variations in surgical skills have been raised in the LACC trial.

Nevertheless, RCT represents the epitome amongst all types of studies in terms of the level of evidence provided. However systematic reviews with meta-analyses are also considered to provide level-1 evidence. When looking specifically at the question of minimally invasive surgery (MIS) in the management of cervix cancer, the results are conflicting as three meta-analyses showed non-inferiority of MIS while one confirmed the results of the LACC trial [10–13].

The LACC trial was not designed to determine the cause of the inferior outcomes observed in the minimally invasive surgery group [14]. In this review, we wish to critically explore the potential causes driving the poorer oncologic outcomes associated with MIS, including the type of MIS surgery (conventional laparoscopy vs. robotic surgery), the size of the lesion ($\leq$2 cm vs. >2 cm), the impact of $CO_2$ pneumoperitoneum, prior conization, the use of a uterine manipulator, the use of protective measures (vaginal closure before colpotomy), and the effect of surgical expertise/the learning curve.

## 2. Surgical Approach

### 2.1. Robotic Surgery vs. Conventional Laparoscopy

One of the major criticisms against the LACC trial was that only 16% of the patients in the minimally invasive arm underwent robotic surgery, which is clearly not reflective of the current practice patterns in the United States or in Nordic European countries where robotic surgery is most often utilized [4]. However, in a large epidemiologic study involving 2461 women from the United States, even when the vast majority of the patients in the MIS group underwent robotic surgery (80% robotic; 20% conventional laparoscopy), MIS was still associated with a higher risk of death compared with open surgery (HR = 1.61; 95% CI: 1.18–2.21) [15].

It is well-known that robotic technology has improved conventional laparoscopy by providing 3-D visual access, instrument articulation, tremor filtration, and motion scaling, resulting in improved dexterity [16,17]. These robotic surgical improvements may influence surgical technique and therefore possibly oncologic outcomes. However, no study compared the radicality of the hysterectomy specimen (size of parametrium or length of vaginal cuff) between robotic and laparoscopic surgery. In addition, only one study, with a small number of patients, compared robotic surgery (*n* = 24) vs. conventional laparoscopy (*n* = 32) in terms of oncologic outcome and found no statistically significant difference between the two techniques, with disease-free survival of 95.8% and 90.6%, respectively [18].

### 2.2. Robotic Surgery vs. Open Surgery

In the study by Yang et al., the authors retrospectively reviewed the Mayo Clinic Cancer Registry for women with cervical cancer who underwent robotic or open radical hysterectomies from 2000 to 2017. A total of 333 patients were included (181 open, 152 robotic). The vaginal margins and the width of the parametrium were measured after pathological re-assessment of the surgical specimens. The authors found no difference between the open and robotic groups in terms of vaginal length (2.1 vs. 2.0 cm, *p* = 0.22) or mean parametrium width on either side (left side 2.5 vs. 2.4 cm, *p* = 0.99, respectively; right side 2.4 vs. 2.4 cm, respectively, *p* = 0.74). Yet, even if the pathological specimens were comparable in terms of specimen radicality, the authors found that robotic radical hysterectomy (RH) had inferior OS and PFS compared to open surgery [19].

These findings suggest that the poor outcomes observed in the robotic group cannot be explained by the radicality of the surgery, but rather possibly by technique-related causes leading to tumor contamination, such as improper tumor handling, peritoneal spillage at the time of colpotomy with $CO_2$ insufflation, or the use of uterine manipulators.

There are currently two ongoing prospective RCTs aiming to exclusively compare robotic surgery to open surgery in early-stage cervical cancer (RACC and ROCC trials) [20,21]. In both trials, the investigators require (ROCC trial) or recommend (RACC trial) tumor-containing methods to minimize contamination, such as performing the colpotomy entirely vaginally after intracorporeal radical dissection is completed or the vaginal mucosal layer is developed and sutured together over the cervix and tumor and then performing the colpotomy intracorporeally. Moreover, in both trials, transcervical/intrauterine manipulators are not permitted, while in the ROCC trial, vaginal manipulators can be used.

### 2.3. Meta-Analysis

(a) MIS (laparoscopy and robotic surgery) vs. OPEN SURGERY

A recently published systematic review with a meta-analysis calculated a pooled hazard of death or recurrence 71% higher among patients who underwent MIS compared with those who underwent open surgery, confirming the results of the LACC trial. This review included both standard and robotic laparoscopy [10] (Table 1).

**Table 1.** Systematic reviews and meta-analyses comparing outcomes of MIS and open radical hysterectomy in the management of cervical cancer tumors.

| Authors | Number of Studies | FIGO Stage (2009) | Number of Patients | Comparison Group | Results |
|---|---|---|---|---|---|
| Nitecki et al. [10] | 15 | IA1 to IIA | Total = 9499<br>CL = 2009<br>RL = 2675<br>Open= 4815 | MIS (Robotic + conventional)<br>vs.<br>Open | The pooled hazard of recurrence or death was 71% higher among patients who underwent minimally invasive radical hysterectomy compared with those who underwent open surgery (HR 1.71; 95% CI, 1.36–2.15; $p < 0.001$). |
| Cao et al. [11] | 22 | IA1 to IIB | Total = 2922<br>CL = 1230<br>Open = 1692 | Conventional laparoscopy<br>vs.<br>Open | No significant differences were found in 5-year DFS and OS (HR = −0.01; 95% CI, −0.08, 0.07; $p = 0.88$). |
| Wang et al. [12] | 12 | IA1 to IIA | Total = 1539<br>CL = 754<br>Open = 785 | Conventional laparoscopy<br>vs.<br>Open | There were no significant differences in 5-year overall survival (HR 0.91, 95% CI 0.48–1.71; $p = 0.76$) and 5-year disease-free survival (HR 0.97, 95% CI 0.56–1.68; $p = 0.91$). |
| Geetha et al. [13] | 47 | NA | Total = 3218<br>CL = 1339<br>RL = 327<br>Open = 1552 | Conventional laparoscopy<br>vs.<br>Robotic laparoscopy<br>vs.<br>Open | The recurrence rate between the three types of radical hysterectomy procedures was similar. |

CL: Conventional laparoscopy. RL: Robotic laparoscopy.

(b) MIS (laparoscopy only) vs. OPEN SURGERY

Only two systematic reviews with meta-analyses exclusively included conventional laparoscopic radical hysterectomies (LRH) in comparison to abdominal radical hysterectomy (ARH). In the study by Cao et al., 2922 patients were included (1230 LRH and 1692 ARH) and no differences in terms of 5-year DFS or 5-year OS were identified. [11] Similarly, in the study by Wang et al., 1539 patients were included (754 LRH and 785 ARH) and there were no significant differences in 5-year OS (HR 0.91, 95% CI 0.48–1.71; *p* = 0.76) and 5-year DFS (hazard ratio [HR] 0.97, 95% CI 0.56–1.68; *p* = 0.91) [12].

(c) Laparoscopy vs. Robotic surgery vs. OPEN SURGERY

The only systematic review with meta-analysis comparing the three techniques, including 21 studies of laparoscopic radical hysterectomy (1339 patients), 14 studies of open radical hysterectomy (1552 patients), and 12 studies of robotic radical hysterectomy (327 patients), found a similar recurrence rate between the three techniques [13].

In summary, the available data would suggest that the radicality of the surgery between robotic and open surgery is comparable, and that the difference in outcomes observed between MIS and open surgery in the LACC trial and in the national cancer database (NCDB) study cannot be blamed exclusively on the use of conventional laparoscopy as both MIS techniques (laparoscopy and robotics) have shown similar outcomes in retrospectives studies and meta-analyses [4–15]. Therefore, besides the surgical technique, other factors are probably involved in the poorer oncologic outcome associated with MIS, such as tumor size, the use of protective measures or not, and prior conization.

## 3. Tumor Size

Tumor size is a recognized prognostic factor in cervical cancer with a strong correlation with the rates of lymph node metastasis and parametrial involvement [22]. Numerous retrospective series have shown that the rate of parametrial infiltration is very low in patients with low-risk disease (lesions < 2 cm with negative nodes and limited stromal invasion) [23], suggesting that these patients may not benefit from radical surgery. The results of the SHAPE-CX5 trial, a prospective randomized trial evaluating whether simple hysterectomy and pelvic node dissection are non-inferior to radical hysterectomy and pelvic node dissection in terms of pelvic relapse-free survival in patients with a tumor size inferior to 2 cm, are awaited [24]. To reflect the trend towards less radical surgery in low-risk disease, the 2018 International Federation of Gynecology and Obstetrics (FIGO) uterine cervical cancer staging system subdivided stage IB disease and introduced a new tumor size cut-off value of ≤2 cm (now stage IB1) [25].

In the LACC trial, even if the two groups (MIS vs. open surgery) were equally balanced regarding tumor size (≥2 cm: open =121; MIS = 123 and <2 cm: open = 147; MIS = 150), the authors concluded that the results of the trial could not be generalized to patients with "low-risk" cervical cancer with a tumor size <2 cm because the trial was not powered to evaluate the oncologic outcomes of the two surgical approaches in that setting [4].

*3.1. Studies That Did Not Confirm That MIS Was Associated with a Worse Outcome in Patients with Lesions ≤ 2 cm*

In a large epidemiologic retrospective national cancer database study in the United States published by Melamed et al., the findings suggest that MIS, regardless of tumor size, was associated with a higher risk of death compared to open surgery among women who underwent radical hysterectomy for early-stage cervical cancer. However, the study was unable to precisely estimate the association between MIS and all-cause mortality among women who had tumors < 2 cm [14].

In a European multicenter retrospective observational cohort study (The SUCCOR study), evaluating DFS in patients with stage IB1 (FIGO 2009) cervical cancer undergoing open vs. minimally invasive radical hysterectomy, the authors found that MIS doubled the risk of relapse. However, in the group of patients with tumors ≤ 2 cm, there was no statistical difference. In addition, the risk of death was 2.42-times higher in the MIS group

compared to the open surgery group. Once again, this difference was only significant in patients with tumors > 2 cm (HR, 2.26; 95% CI, 1.18 to 4.36; *p* = 0.014) but not in patients with tumors ≤ 2 cm (HR, 2.77; 95% CI, 0.91 to 8.47; *p* = 0.072) [26].

In a nation-based retrospective study from the Netherlands, 384 patients with cervix cancer with clinical tumor size ≤ 2 cm were included (LRH = 166; ARH = 218). The authors found no difference in the 5-year DFS (91.4% and 96.0% in the ARH and LRH groups, respectively (0.44 [0.16–1.27])), and the 5-year OS (96.4% and 98.5% (0.39 [0.08–1.86])) [27].

In a large Chinese study (*n* =1852), only tumors < 2 cm were examined, and similar oncologic outcomes were observed between the laparoscopy and laparotomy groups (OS 96.3% vs. 96.6%; *p* = 0.692; DFS 92.6% vs. 94.9%; *p* = 0.064) [28]

Lastly, Kim et al. conducted a retrospective matched cohort study of patients in two high-volume tertiary institutional hospitals in Korea and found that MIS was associated with a significantly higher recurrence rate than open surgery. However, this association was not observed among matched stage IB1 patients with tumor size ≤ 2 cm on pre-operative MRI [29].

*3.2. Studies Who Did Confirm That MIS Was Associated with a Worse Outcome Compared with Open Surgery for Patients Even with Tumor Size ≤ 2 cm*

In the Korean Gynecologic Oncology Group Study (KGOG 1028), 248 patients with a tumor size under 2 cm and without adjuvant treatment (low-risk early-stage cervical cancer patients) were included. The MIS approach (62 patients) showed poorer DFS when compared to the open approach (186 patients) specifically for pelvic and hematogenous recurrences [30].

In an additional Chinese study exclusively including patients with stage IB1 disease with a clinical and MRI tumor size ≤ 2 cm, the authors found that patients in the MIS group had worse 5-year DFS compared to those in the open group (90.4% vs. 97.7%; *p* = 0.02). However, no significant difference in 5-year OS between the two groups was demonstrated (96.9% vs. 99.4%, *p* = 0.33). The authors explained this difference by the fact that some of the patients with recurrences in the MIS group might have been salvaged with either chemoradiation or chemotherapy alone [31].

Uppal et al., in a large multi-institutional retrospective study from the US, reported 264 patients with tumors ≤ 2 cm on final pathology. After excluding those with no residual tumor on final pathology, the MIS approach was noted to be independently associated with a higher likelihood of recurrence (aHR, 6.31; 95% CI, 1.24 to 31.9) [32].

Finally, Odetto et al. reported a recurrence rate in patients with tumor size ≤ 2 cm as high as 12% (7/58) when looking specifically at patients with early-stage cervical cancer who underwent a laparoscopic radical hysterectomy in Argentina [33].

In summary, it is difficult to draw conclusions with regards to tumor size as a possible explanation for the poorer outcomes associated with MIS. The results of the retrospective studies are conflicting, with significant differences in the median follow-up time between the open surgery groups and the laparoscopic groups in some studies. In addition, the cut-off measure of 2 cm was determined differently from one study to another. In some studies, tumor size was determined by clinical examination, while in others, MRI measurements or even final pathology measurements were used (Table 2).

It is probably not the tumor size itself but possibly the presence or not of a macroscopic tumor that drives the poorer outcomes in patients who underwent MIS. Patients with a tumor size under 2 cm represent a heterogeneous group, including patients with a post-cone microscopic tumor or no residual tumor. Nevertheless, potential explanations for the poorer results associated with MIS in cervix cancer, even with a tumor size under 2 cm, are still a subject of debate. Hypotheses include the potential increased tumor spillage attributable to the use of a uterine manipulator or exposure of tumor cells to the peritoneal cavity when an intracorporeal colpotomy is performed or tumor cell growth and spread enhanced using $CO_2$ gas insufflation when MIS is employed.

**Table 2.** Analyses from studies comparing outcomes of MIS and open radical hysterectomy in the management of cervical cancer tumors $\leq 2$ cm.

| Authors | Inclusion Criteria (Stage According to FIGO 2009) | Number of Patients | Follow-Up (Months) Median | Results |
|---|---|---|---|---|
| Chen C et al. [28] | IB1; ≤2 cm Tumor size: Final pathology | Total = 1852 MIS = 926 Open = 926 | 36 | - Comparable survival outcomes were observed between the laparoscopic and abdominal groups (OS 96.3% vs. 96.6%; *p* = 0.692; DFS 92.6% vs. 94.9%; *p* = 0.064) |
| Paik et al. [30] KGOG 1028 study | IB1, IIA1; ≤2 cm Tumor size: Clinical | Total = 248 MIS = 62 Open = 186 | 69.1 (range: 3.0–173.3) | - No significant difference in OS between the two groups (*p* = 0.562, HR not calculated).<br>- Patients treated with laparoscopy showed inferior DFS (HR 12.987 [95% CI 1.451–116.244], *p* = 0.003) |
| Chiva et al. [26] SUCCOR study | IB1 Tumor size: MRI | Total = 303 MIS = 151 Open = 152 | 59 (range: 1–83) | - No difference in risk of recurrence (HR, 1.63; 95% CI, 0.79 to 3.40; *p* = 0.19)<br>- No difference in risk of death (HR, 2.77; 95% CI, 0.91 to 8.47; *p* = 0.072) |
| Wenzel et al. [27] Deutsch study | IA2 LVSI+; IB1; IA2 Tumor size: Clinical | Total = 384 MIS = 166 Open = 218 | DFS = 35 (range: 0–100) OS = 56 (range: 1–109) | - No difference in oncologic outcome: 5-year DFS 91.4% and 96.0% in the Open and MIS group, respectively (0.44 [0.16–1.27]). Five-year OS was 96.4% and 98.5% (0.39 [0.08–1.86]) |
| Kim et al. [29] Korean study | IB1; ≤2 cm Tumor size: MRI | Total = 246 MIS = 125 Open = 121 | 66.2 | - Both groups showed similar OS (5-year: 96.4% vs. 98.6%; *p* = 0.6) and PFS (3-year: 93.1% vs. 90.0%; *p* = 0.8) |
| Uppal et al. [32] | IA1, IB1 Tumor size: Final pathology | Total = 264 MIS = 182 Open = 82 | MIS = 30.7 (range: 13.75–51.44) Open = 44.6 (range: 20.98–67.39) | - No significant difference in OS and DFS.<br>- But when excluding patients with no residual tumor on final pathology, the MIS approach was noted to be independently associated with a higher likelihood of recurrence (aHR, 6.31; 95% CI, 1.24 to 31.9 |
| Odetto et al. [33] | IA1 LVSI+, IA2, IB1 Tumor size: MRI | MIS = 58 | 39 (range: 11–83) | The recurrence rate in tumors ≤ 2 cm was 12%. |

**Table 2.** *Cont.*

| Authors | Inclusion Criteria (Stage According to FIGO 2009) | Number of Patients | Follow-Up (Months) Median | Results |
|---|---|---|---|---|
| Chen X et al. [31] | IB1; ≤2 cm Tumor size: Clinical + MRI | Total = 325 MIS = 129 Open = 196 | MIS = 51.8 (range: 2–115) Open = 49.5 (range 3–108) | - No significant difference in 5-year OS between the groups (96.9% vs. 99.4%; $p = 0.33$)<br>- Worse 5-year DFS in the MIS group compared to the open surgery group (90.4% vs. 97.7%; $p = 0.02$)<br>- Patients who underwent open surgery and MIS had recurrence rates of 2.3% and 9.6%, respectively |

## 4. The Impact of $CO_2$ Pneumoperitoneum

At least four cases of intraperitoneal dissemination after laparoscopic radical hysterectomy for cervical cancer have been reported between 1997 and 2012 [34–36].

Considering these findings, Kong et al. conducted a retrospective analysis of 128 patients with FIGO stage IB and IIA cervical cancer treated with MIS between 2006 and 2013 [37]. Their results showed that disease recurrence was higher in the group where an intracorporeal colpotomy was performed in comparison with the group where a vaginal colpotomy was used (16.3% vs. 5.1%, $p = 0.057$). The authors incriminated $CO_2$ as the main factor for these results. They concluded that, contrary to vaginal colpotomy, intracorporeal colpotomy performed under pneumoperitoneum pressure and promoted by circulating $CO_2$ might be associated with an increased risk of intraperitoneal tumor spillage.

Several theories could explain the role of $CO_2$ in carcinogenesis.

In an experimental animal study, Volz et al. suggested that intraperitoneal tumor spread may be connected to the presentation of cancerous tumor cells to the circulating $CO_2$ pneumoperitoneum associated with the disturbance of the superficial mesothelial layer by high $CO_2$ pressure, leading to tumor cell implantation [38].

Several in vitro studies showed that the cell colony formation increased significantly after stimulation by $CO_2$ [39,40]. However, the specific mechanism supporting this effect is not yet precisely determined. Some pathways have been studied, such as the influence of $CO_2$ on the transition of cells from G1 to S phase [41], or the acidic environment of $CO_2$ that could activate mitosis enzymatic activity promoting tumor growth [42].

In addition, studies suggest that the acidosis of the intraabdominal environment suppresses the peritoneal immune system [43]. The acidic environment also induces damage to the peritoneum, making the adhesion and implantation of tumor cells easier [38,44].

One of the solutions to overcome the possible negative consequences of $CO_2$ could be to use low-pressure pneumoperitoneum combined with an abdominal wall lift [45]. Moreover, as mentioned previously, performing vaginal colpotomy instead of intracorporeal colpotomy could minimize the potential deleterious effect of $CO_2$ in addition to decreasing tumor manipulation and tumor cell contact with peritoneal surfaces.

## 5. Conization

Manipulation of macroscopic tumors during the surgery and tumor leakage could play a substantial role in the increased risk of recurrence associated with MIS. Thus, it is plausible that patients with a preoperative diagnosis confirmed by a cervical biopsy only may possibly have a higher risk of intra-abdominal tumor exposure during surgery compared to patients having had diagnostic conization.

Assuming this hypothesis, Casarin et al. performed a multicenter retrospective study comparing patients having an open radical hysterectomy with patients having MIS surgery

(IA1 (LVSI+), IA2, IB1) with all the procedures performed without vaginal closure or tumor exclusion before the colpotomy and with the use of a uterine manipulator [46]. The authors found that the recurrence rate in patients who underwent preoperative conization was 1.1% (1/93) vs. 16.1% (15/93) for those who had a preoperative cervical biopsy only ($p < 0.001$). The association was also confirmed in the FIGO 2009 stage IB1 sub-analysis (recurrence rate: Conization 1.8% vs. cervical biopsy 17.2%; $p = 0.004$). The 5-year Kaplan–Meier curves showed preoperative conization (vs. cervical biopsy) to be associated with better disease-free survival (hazard ratio 0.12; 95% confidence interval 0.04–0.33). Even in the subgroup of patients with FIGO 2009 IB1 tumor $\geq$2 cm, the analysis showed a reduction by approximately 15% of the risk of recurrence in patients who had preoperative conization (vs. cervical biopsy).

Similarly, in a recent study, the SUCCOR group evaluated (the SUCCOR CONE STUDY) the role of conization as a protective factor for patients undergoing a radical hysterectomy. After homogenizing the population using the Propensity Matching Score (PMS), they showed that preoperative conization could have a protective effect, with the risk of relapse reduced by 65% (HR: 0.35 CI 95% (0.16–0.75) $p = 0.007$) and the risk of death reduced by 75% (HR: 0.25 95% CI (0.07–0.90) $p = 0.033$) [47].

In another recent study aiming to identify the role of cervical conization before radical hysterectomy (RH), the authors identified low-risk, early-stage cervical cancer patients (parametria-negative, node-negative, margin-negative) who received primary Type C RH at two Korean institutions. Their results showed that the conization group had a significantly better DFS than the no-conization group (3-year DFS rate, 94.2% vs. 86.3%; $p = 0.012$), but similar OS. Among the open RH patients ($n = 96$), no difference in DFS was observed between the conization and control groups ($p = 0.984$). In contrast, amongst the MIS RH patients ($n = 192$), the conization group showed significantly better DFS vs. the control (3-year DFS rate, 95.7% vs. 82.9%; $p = 0.005$) [48].

Finally, in a large multi-institutional retrospective review study, prior conization was associated with a significantly lower risk of recurrence: 4.9% (26/535) vs. 16.2% (43/266; $p = 0.001$). [32]. The authors also specifically analyzed the outcomes of 243 patients who underwent conization and had **no residual tumor on the preoperative assessment before radical hysterectomy**. In this particular group of patients, the rate of recurrence was 1.4% for the open group and 2.9% for the MIS group ($p = 0.48$). They also found no recurrences in patients with **no residual tumor on their final pathology** ($n = 222$; open = 53; MIS = 169) independently from the initial FIGO clinical stage determined preoperatively.

Even if the results of these four studies are concordant and suggest the "protective" effect of preoperative conization on the risk of cancer recurrence, the findings cannot be generalized. Moreover, most of the patients who had preoperative conization underwent their surgery in non-referral centers, and for this reason, it is difficult to gain access to the pathological report, including information on the presence/absence of disease at the margins of the cone specimen. Nevertheless, it would intuitively make sense that diagnostic conization and the absence of a macroscopic tumor on the cervix could possibly be associated with a risk reduction of tumor spillage and recurrence at the time of radical hysterectomy.

## 6. Uterine Manipulator and Vaginal Closure

Fundamental rules of oncologic surgery include the avoidance of tumor spillage and tumor manipulation and resection in tumor-free margins [49]. Techniques of MIS radical hysterectomy commonly include the use of a uterine manipulator even in patients with a visible cervical tumor, which, consequently, likely violates the above principles.

### 6.1. Vaginal Closure

Kohler et al. published a series of 389 patients with early-stage cervical cancer undergoing vaginally assisted laparoscopic radical hysterectomy or laparoscopic-assisted radical vaginal hysterectomy. They used protective measures with the rigorous avoidance of any uterine manipulator, the creation of a vaginal cuff covering the tumor, and finally, by

performing an extraperitoneal colpotomy. They reported oncologic outcomes similar to those of the open radical hysterectomy arm reported in the LACC trial [50]. After a median follow-up of 99 months (range 1–288), the 3-, 4.5-, and 10-year disease-free survival rates were 96.8%, 95.8%, and 93.1%, and the 3-, 4.5-, and 10-year overall survival rates were 98.5%, 97.8%, and 95.8%, respectively.

The impact of vaginal closure was also evaluated in the SUCCOR study. DFS at 4.5 years was significatively higher in the group with protective vaginal closure than in the group without, at 93% and 74% ($p < 0.001$), respectively. MIS patients without protective vaginal closure had more than doubled their risk of recurrence compared to the open approach (HR, 2.58; 95% CI, 1.70 to 3.95; $p < 0.001$). Patients who underwent MIS with vaginal closure had comparable relapse rates compared to those who underwent open surgery (HR, 0.63; 95% CI, 0.15 to 2.59; $p < 0.52$). In terms of survival, MIS patients without vaginal closure had a 2.85-times-higher risk of death when compared with those who underwent an open approach ($p < 0.001$) [26].

The above data would suggest that protective measures are associated with lower risks of cancer recurrence following MIS surgery in patients with cervical cancer.

*6.2. Uterine Manipulator*

Another sub-analysis of the SUCCOR study specifically explored the impact of the uterine manipulator in MIS. In the group of patients where a uterine manipulator was used, there were more relapses (26.3%) in comparison with the group without the use of a uterine manipulator (16%). The DFS at 4.5 years was 73% in the uterine manipulator group and 83% in the group without ($p = 0.0001$).

In the group of patients with tumors >2 cm, the adverse effect of the uterine manipulator was statistically significant in terms of risk of relapse (HR, 3.05; 95% CI, 1.73 to 5.38; $p < 0.001$) while its use in tumors $\leq$ 2 cm did not show statistical differences (HR, 2.25; 95% CI, 0.96 to 5.26; $p = 0.06$). The use of a uterine manipulator also adversely impacted OS in patients who underwent MIS (HR, 3.00; 95% CI, 1.60 to 5.63; $p = 0.001$) [26]. Patients who underwent MIS **with** a uterine manipulator had a 2.76-times-higher chance of relapse compared with those in the open approach. However, patients undergoing minimally invasive surgery **without** the uterine manipulator had similar rates of relapse compared to those who underwent open surgery (HR, 1.58; 95% CI, 0.79 to 3.15; $p = 0.20$) [26].

Conversely, in another study focusing on the role of the uterine manipulator, the authors identified a total of 224 patients undergoing MIS (171 were laparoscopic and 53 were robotic); 115 had surgery with the use of an intra-uterine manipulator while 109 did not. After controlling for confounding factors (positive margins or parametria, lymph node metastasis, the presence of residual tumor at hysterectomy, and tumor size), the use of an intra-uterine manipulator was no longer significantly associated with worse recurrence-free survival (HR 0.4, 95% CI 0.2 to 1.0, $p = 0.05$) [51]. The authors found that tumor size is a significant predictor of disease recurrence in patients who underwent an MIS radical hysterectomy, indicating possibly that the key mechanism associated with cancer recurrence is the intra-abdominal spreading of malignant cells during a minimally invasive intracorporeal colpotomy.

Even if the biology and surgical anatomy of endometrial cancer differ considerably from cervical cancer, a large retrospective multicentric study with a total of 2661 women found that the use of uterine manipulators in early-stage endometrial cancer operated by MIS was associated with lower DFS, a higher recurrence rate, and higher risk of death than the same surgery performed without the use of uterine manipulators [52].

Therefore, even if endometrial cancer is considered an organ-confined disease, with the myometrium acting as a containment barrier, the uterine device may explain the alteration of the myometrial barrier with macroscopic injuries secondary to uterine perforation and spread of the tumor into the peritoneal cavity. Another hypothesis is the microscopic pathway of dissemination by increasing intrauterine pressure and pushing tumor cells to go beyond the myometrial barrier, spreading outside the uterus cavity by a passive

effect across the fallopian tubes and lymphovascular space [53]. The same may occur in cervical cancer. Placement of the uterine manipulator, particularly if there is a macroscopic disease in the cervix, may lead to tumor cell disruption and implantation at the vaginal vault or intraperitoneally.

### 7. The Effect of Surgical Expertise/Learning Curve

Beyond the technical and anatomical aspects previously discussed, the human factor characterized by surgical proficiency could be one of the explanations driving the unexpected poor oncologic results of the MIS arm in the LACC trial.

It is well-known that surgeons' MIS technical skills and abilities are significantly associated with oncologic outcomes in pelvic tumors such as colorectal cancer [54].

The number of cases performed (surgical volume) is one of the factors that unquestionably impact surgical efficiency [55,56].

One of the criticisms against the LACC trial was that the participating surgeons were required to provide only 10 documented minimally invasive radical hysterectomies and 2 unedited videos as evidence of technical competency [4].

Interestingly, in a study looking specifically at the impact of the learning curve in the surgical management of cervical cancer, the authors found that even with 10 cases performed, surgeons are still in the early phase of the learning curve, which is associated with a poorer prognosis [57]. Indeed, the authors found that PFS was significantly poorer in the early phase compared with the late phase (5-year progression-free rate, 100% and 78.2%, respectively, $p = 0.014$). They concluded that the minimum number of cases required to achieve surgical proficiency was 13 for the laparoscopic radical hysterectomy and 21 for the robotic radical hysterectomy.

The number of MIS cases from which there is a reduction in the risk of recurrence varies from one study to another, ranging from 19 in the study by Pedestrian et al. [58] to as high as 61 in the study by Baeten et al. [59], and well above the 10 cases required in the LACC trial.

In another study, the institutional learning curve for robotic radical hysterectomies, represented by two consecutive time periods (2006–2012 vs. 2013–2018), was one of the significant predictors for PFS (HR 0.065, $p = 0.0162$) in the multivariate analysis. Moreover, in the same study, the authors found that even if PFS was significantly different between the abdominal radical hysterectomy group and the overall robotic radical hysterectomy group ($p = 0.002$), there was no difference in PFS between the two surgical approaches when the surgery was performed in the second time-period ($p = 0.629$) [60].

Yang et al. also studied the impact of surgeons' proficiency on the survival impact of MIS surgery in early-stage cervical cancer. They categorized patients who underwent surgery into four phases (phase 1, 1–10 cases; phase 2, 11–20 cases; phase 3, 21–30 cases; phase 4, more than 30 cases). The authors found that when stratified by surgical phases, the OS and DFS of the MIS group in the phase 1 group were significantly lower compared to the other phases and compared to the open surgery group after adjusting for age, BMI, FIGO stage, histologic subtype, and grade. Interestingly, when looking specifically at patients in the phase 1 group, the OS and DFS were in line with those of the LACC trial [61].

In summary, surgical proficiency and surgical volume seem to play a substantial role in the oncologic outcomes of patients with early-stage cervical cancer operated by MIS. However, the reasons behind these findings are difficult to identify. Nevertheless, the radicality of the surgery as well as technical factors such as the avoidance of tumor spillage and tumor manipulation probably improve with surgical experience.

### 8. Conclusions

The core of scientific discovery is the principle of repeated testing of a hypothesis. Additional RCTs addressing the concerns of the LACC trial are warranted, justified with equipoise, and already underway [20,21].

Future trials evaluating the safety of minimally invasive radical hysterectomies should focus on modification of the surgical technique and careful selection of patients with smaller tumors, prior conization, and minimal tumor manipulation using protective measures such as vaginal closure and avoiding uterine manipulators.

Recently, the ESGO developed a list of quality indicators for surgical treatment of cervical cancer that can be used to audit and improve clinical practice [62].

Recent data from the United States indicate that since the publication of the LACC trial in 2018, there has been a 63% decrease in the rate of radical hysterectomies performed by MIS, but at the same time, a 23% increase in the rate of postoperative complications associated with open surgery [63].

However, two publications following the LACC trial and based on the same randomized population showed no difference in the overall incidence of intraoperative or postoperative adverse events between minimally invasive and open radical hysterectomy for early cervical cancer and found a similar postoperative quality of life between the two surgical approaches [64,65].

In the future, it is hoped that a compromise can be reached to better identify patients who could still safely benefit from MIS surgery in order to reduce surgical morbidity. For now, open surgery should be considered the standard-of-care surgical approach.

**Author Contributions:** O.T.: Conceptualization, methodology, literature review, writing—original draft preparation; M.P.: Writing—review and editing. All authors have read and agreed to the published version of the manuscript.

**Funding:** This research received no external funding.

**Institutional Review Board Statement:** Not Applicable.

**Informed Consent Statement:** Not Applicable.

**Conflicts of Interest:** The authors declare no conflict of interest.

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
