# Peer review of "Minimally Invasive Surgery for Cervical Cancer in Light of the LACC Trial: What Have We Learned?"

_curroncol, doi:10.3390/curroncol29020093_

Round 1

Reviewer 1 Report

The authors tried to explore the potential causes of poorer oncologic outcomes associated with MIS radical hysterectomry for early stage cervical cancer in the LACC trial through a thorough literature review.

Based on the evidence, the review went very well.

There are some key points to add to this review.

  1. One of the causes of frequent recurs in the MIS group of the LACC trial is the inappropriate surgeon proficeicy criteria. Although MIS RH is a very difficult operation and the learning curve is quite long, the surgeon was allowed to participate in the study even if only 10 cases of MIS RH were performed. With this level of experience, an appropriate immediate surgical outcome cannot be obtained, and an appropriate concologic outcome cannot be obtained. In the current literature, there are several good papers showing the effect of the learning curve on immediate surgical outcome and oncologic outocme in MIS RH. It is necessary to review this in depth and cite these papers.
  2. In MIS RH, in addition to tumor closure and vaginal colpotomy, it is important to change the patient's posistion from the trendelenburg down position to the supine position before colpotomry to prevent transperitoneal tumor spread by gravity. It is also necessary to wash the tumor debri with saline irrigation prior to colpotomy. And, vaginal colpotomy is better than intracorporeal colpotomy. Vaginal colpotomy can be performed by pulling the uterine cervix out of the vaginal orifice, further reducing the possibility of transperitoneal tumor seeding. It is necessary to further describe these additional protective measures.
  3. The authors reported that the use of a uterine manipulator had an adverse effect on the oncologic outcome after MIS RH. However, this is only applicable if the tumor is broken during surgery and protective measures are not taken during colpotomy. Therefore, if tubal clmaping is performed before surgery, the patient's position is changed from the trendelenberg position to the supine position before colpotomy, the broken tumor is washed with saline irrigation, and vagianl colpotomy is performed, then the use of a uterine manipuatior is not a problem. This is the same even when the tumor size is large.

Author Response

Chicoutimi 03/02/2022

Dear Editor and reviewers,

We would like to thank you for your comments and revision. 

Reviewers' comments:

Reviewer #1:

The authors tried to explore the potential causes of poorer oncologic outcomes associated with MIS radical hysterectomy for early-stage cervical cancer in the LACC trial through a thorough literature review.

Based on the evidence, the review went very well.

There are some key points to add to this review.

Comments 1:

One of the causes of frequent recurs in the MIS group of the LACC trial is the inappropriate surgeon proficiency criteria. Although MIS RH is a very difficult operation and the learning curve is quite long, the surgeon was allowed to participate in the study even if only 10 cases of MIS RH were performed. With this level of experience, an appropriate immediate surgical outcome cannot be obtained, and an appropriate oncologic outcome cannot be obtained. In the current literature, there are several good papers showing the effect of the learning curve on immediate surgical outcomes and oncologic outcomes in MIS RH. It is necessary to review this in depth and cite these papers.

Response 1:

A paragraph (number 7) focusing on the effect of surgical expertise and learning curve on oncologic outcome has been added

Comments 2:

In MIS RH, in addition to tumor closure and vaginal colpotomy, it is important to change the patient's position from the Trendelenburg down position to the supine position before colpotomy to prevent transperitoneal tumor spread by gravity. It is also necessary to wash the tumor debris with saline irrigation prior to colpotomy. And, vaginal colpotomy is better than intracorporeal colpotomy. Vaginal colpotomy can be performed by pulling the uterine cervix out of the vaginal orifice, further reducing the possibility of transperitoneal tumor seeding. It is necessary to further describe these additional protective measures.

Response 2:

Evidence from a study showing better oncologic outcomes in patients having vaginal colpotomy in comparison with patients having intracorporeal colpotomy has been added (paragraph 4: The impact of CO2 pneumoperitoneum)

The other remarks although being logical and making sense, are not based on strong evidence: change the patient's position from the Trendelenburg down position to the supine position before colpotomy; washing the tumor debris with saline irrigation prior to colpotomy.

Comments 3:

The authors reported that the use of a uterine manipulator had an adverse effect on the oncologic outcome after MIS RH. However, this is only applicable if the tumor is broken during surgery and protective measures are not taken during colpotomy. Therefore, if tubal clamping is performed before surgery, the patient's position is changed from the Trendelenburg position to the supine position before colpotomy, the broken tumor is washed with saline irrigation, and vaginal colpotomy is performed, then the use of a uterine manipulator is not a problem. This is the same even when the tumor size is large.

Responses 3:

Even if the data regarding the impact of the uterine manipulator on oncologic outcomes are conflicting, there is no evidence that tubal clamping, changing position before colpotomy, or washing with saline irrigation could have an impact on the potentially deleterious impact of the uterine manipulator.

We hope that the changes made to the manuscript will be to the satisfaction of the Editor and reviewers and that our revised manuscript will be accepted for publication.

With kind regards,

Dr Omar Touhami

Gynecologic Oncology Division, Department of Obstetrics and Gynecology, Centre intégré universitaire de santé et services sociaux CIUSSS du Saguenay—Lac-Saint-Jean, Sherbrooke University, Québec, Canada

Reviewer 2 Report

The article is well written, ordered and addresses the problem in a correct way.

Feedback:

In a recently published paper by our Spanish-Portuguese group of robotic surgery, (our group sent data to SUCCOR study), we concluded that in addition to the risk factors for relapse mentioned by the authors in this review, we found, in addition to the tumor size greater than 20 mm, the histological subtype of adenocarcinoma, tumor grades 2 and 3, the presence of positive pelvic nodes and the non-performance of selective sentinel node biopsy.

Likewise, the need for the use of surgical quality indicators (QI) advocated by the ESGO for minimally invasive surgery is reinforced.

I believe that some reflection on these data should be included in the manuscript and any comments on the influence of QI on the choice of the minimally invasive surgery.

Cancers 2020, 12, 3387; doi:10.3390/cancers12113387

Author Response

Chicoutimi 03/02/2022

Dear Editor and reviewers,

We would like to thank you for your comments and revision. 

Reviewers' comments:

Reviewer #2:

The article is well written, ordered, and addresses the problem in a correct way.

Comments 1:

In a recently published paper by our Spanish-Portuguese group of robotic surgery, (our group sent data to SUCCOR study), we concluded that in addition to the risk factors for relapse mentioned by the authors in this review, we found, in addition to the tumor size greater than 20 mm, the histological subtype of adenocarcinoma, tumor grades 2 and 3, the presence of positive pelvic nodes and the non-performance of selective sentinel node biopsy.

Likewise, the need for the use of surgical quality indicators (QI) advocated by the ESGO for minimally invasive surgery is reinforced.

I believe that some reflection on these data should be included in the manuscript and any comments on the influence of QI on the choice of minimally invasive surgery.

Cancers 2020, 12, 3387; doi:10.3390/cancers12113387

Response 1:

We acknowledge the opinion of the reviewer, in fact, the paper published by the Spanish-Portuguese group of robotic surgery is very interesting.

However, our paragraph on tumor size focuses only on studies including specifically tumors with sizes under 2 cm.

The paper on the ESGO surgical quality indicators (QI) is also very interesting. A reference to this publication has been made in our conclusion.

We hope that the changes made to the manuscript will be to the satisfaction of the Editor and reviewers and that our revised manuscript will be accepted for publication.

With kind regards,

Dr Omar Touhami

Gynecologic Oncology Division, Department of Obstetrics and Gynecology, Centre intégré universitaire de santé et services sociaux CIUSSS du Saguenay—Lac-Saint-Jean, Sherbrooke University, Québec, Canada

Reviewer 3 Report

The paper is of high actuality and importance for the everyday practice.
I acknowledge the tables, which are informative.
Despite several strengths, the paper does not address some aspect, which 
would shed more light on the results of the LACC results.
For a better interpretation of the LACC results, I would suggest to add 
paragraphs addressing following aspects: the impact on quality of life, the 
role of surgical volume, and the role of CO2 pneumoperitoneum.
Similarly, two key "spin-off" papers to the LACC trial (authored by the LACC 
investigators) are not included (and not discussed).

My detailed recommendations:

1) Please, address the most important reports based on the same LACC trial, 
which indicated two further dimensions to be considered, namely quality of 
life and surgical complications.
The data of LACC trial provided evidence, that - against expectations - the 
open approach did not result in higher complication rates and the QoL was 
not inferior as compared to MIS.

Frumovitz M, Obermair A, Coleman RL, et al. Quality of life in patients with 
cervical cancer after open versus minimally
invasive radical hysterectomy (LACC): a secondary outcome of a multicentre, 
randomised, open-label, phase 3, non-inferiority trial. Lancet Oncol. 2020
Jun;21(6):851-860. doi: 10.1016/S1470-2045(20)30081-4.

Obermair A, Asher R, Pareja R, et al. Incidence of adverse events in 
minimally invasive vs open radical hysterectomy in
early cervical cancer: results of a randomized controlled trial. Am J Obstet 
Gynecol. 2020 Mar;222(3):249.e1-249.e10. doi: 10.1016/j.ajog.2019.09.036.
Epub 2019 Oct 3.

2) I miss some recent studies complementing the LACC trial, also in terms of 
the impact of tumor size. Please, include and discuss them:

a) Melamed A, Margul DJ, Chen L, et al. Survival after Minimally Invasive 
Radical Hysterectomy for
Early-Stage Cervical Cancer. N Engl J Med. 2018 Nov 15;379(20):1905-1914.
doi: 10.1056/NEJMoa1804923.

b) Chen X, Yu J, Zhao H, Hu Y, Zhu H. Laparoscopic Radical Hysterectomy 
Results
in Higher Recurrence Rate Versus Open Abdominal Surgery for Stage IB1
Cervical Cancer Patients With Tumor Size Less Than 2 Centimeter: A
Retrospective Propensity Score-Matched Study. Front Oncol. 2021 Jun
10;11:683231. doi: 10.3389/fonc.2021.683231.

3) An important aspect has been completely omitted by the authors: the 
impact
of CO2 pneumoperitoneum on cervical cancer growth.

The chemical properties of COs as well as the physical properties of
pneumoperitoneum (with or without CO2) should be separately considered:
Some useful information can be obtained from:
a) Lv H, Zhou T, Rong F. Proteomic analysis of the influence of CO2
pneumoperitoneum in cervical cancer cells. J Cancer Res Ther. 2021
Nov;17(5):1253-1260. doi: 10.4103/jcrt.jcrt_638_21.
b) Lin F, Pan L, Li L, et al. Effects of a simulated CO2 pneumoperitoneum
environment on the proliferation, apoptosis, and metastasis of cervical
cancer cells in vitro. Med Sci Monit. 2014 Dec 1;20:2497-503. doi:
10.12659/MSM.891179.
c) Kong TW, Chang SJ, Piao X, et al. Patterns
of recurrence and survival after abdominal versus laparoscopic/robotic
radical hysterectomy in patients with early cervical cancer. J Obstet
Gynaecol Res. 2016 Jan;42(1):77-86. doi: 10.1111/jog.12840.

4) The effect of surgical volume/ surgical experience. It was a 
controversial aspect of the LACC trial.

Please, discuss the role of surgical experience on the oncological outcomes 
in LACC/ in MIS.

Yang Y, Huang Y, Li Z. The Surgeon's Proficiency Affected Survival Outcomes
of Minimally Invasive Surgery for Early-Stage Cervical Cancer: A
Retrospective Study of 851 Patients. Front Oncol. 2021 Nov 16;11:787198.
doi: 10.3389/fonc.2021.787198.

5) One minor remark. The format of the reference list is other than the 
format of the mdpi-journals. Moreover, the authors use very different 
formats for references within the same Reference list.

Look at e.g. Ref.13,4,15,16 etc. Any of them applies another citation 
format. It looks a little bit chaotic. 
Please, use a template or a reference 
manager and re-format the reference list accordingly.

Author Response

Chicoutimi 03/02/2022

Dear Editor and reviewers,

We would like to thank you for your comments and revision. 

Reviewers' comments:

Reviewer #3:

The paper is of high actuality and importance for everyday practice. I acknowledge the tables, which are informative. Despite several strengths, the paper does not address some aspects, which would shed more light on the results of the LACC results. For a better interpretation of the LACC results, I would suggest adding paragraphs addressing the following aspects: the impact on quality of life, the role of surgical volume, and the role of CO2 pneumoperitoneum. Similarly, two key "spin-off" papers to the LACC trial (authored by the LACC investigators) are not included (and not discussed).

Comments 1:

Please, address the most important reports based on the same LACC trial, which indicated two further dimensions to be considered, namely quality of life and surgical complications.

The data of LACC trial provided evidence, that - against expectations – the open approach did not result in higher complication rates and the QoL was not inferior as compared to MIS.

Frumovitz M, Obermair A, Coleman RL, et al. Quality of life in patients with cervical cancer after open versus minimally invasive radical hysterectomy (LACC): a secondary outcome of a multicentre, randomized, open-label, phase 3, non-inferiority trial. Lancet Oncol. 2020 Jun;21(6):851-860. doi: 10.1016/S1470-2045(20)30081-4.

Obermair A, Asher R, Pareja R, et al. Incidence of adverse events in minimally invasive vs open radical hysterectomy in early cervical cancer: results of a randomized controlled trial. Am J Obstet Gynecol. 2020 Mar;222(3):249.e1-249.e10. doi: 10.1016/j.ajog.2019.09.036. Epub 2019 Oct 3.

Response 1:

We acknowledge the opinion of the reviewer; however, our paper is focusing on the possible explanations of the poorer oncologic outcomes of MIS.

A reference to both papers on QoL and surgical complications suggested by the reviewers has been added in our conclusion.

Comments 2:

I miss some recent studies complementing the LACC trial, also in terms of the impact of tumor size. Please, include and discuss them:

a) Melamed A, Margul DJ, Chen L, et al. Survival after Minimally Invasive Radical Hysterectomy for Early-Stage Cervical Cancer. N Engl J Med. 2018 Nov 15;379(20):1905-1914. doi: 10.1056/NEJMoa1804923.

b) Chen X, Yu J, Zhao H, Hu Y, Zhu H. Laparoscopic Radical Hysterectomy Results in Higher Recurrence Rate Versus Open Abdominal Surgery for Stage IB1 Cervical Cancer Patients With Tumor Size Less Than 2 Centimeter: A Retrospective Propensity Score-Matched Study. Front Oncol. 2021 Jun 10;11:683231. doi: 10.3389/fonc.2021.683231.

Response 2:

Publication (a): by Melmaed et al. has already been cited and discussed several times in our original submission: reference 15: paragraph 2.1 and 2.3

Comments 3:

An important aspect has been completely omitted by the authors: the Impact of CO2 pneumoperitoneum on cervical cancer growth. The chemical properties of COs as well as the physical properties of pneumoperitoneum (with or without CO2) should be separately considered:

Some useful information can be obtained from:

a) Lv H, Zhou T, Rong F. Proteomic analysis of the influence of CO2 pneumoperitoneum in cervical cancer cells. J Cancer Res Ther. 2021 Nov;17(5):1253-1260. doi: 10.4103/jcrt.jcrt_638_21.

b) Lin F, Pan L, Li L, et al. Effects of a simulated CO2 pneumoperitoneum environment on the proliferation, apoptosis, and metastasis of cervical cancer cells in vitro. Med Sci Monit. 2014 Dec 1;20:2497-503. doi: 10.12659/MSM.891179.

c) Kong TW, Chang SJ, Piao X, et al. Patterns of recurrence and survival after abdominal versus laparoscopic/robotic radical hysterectomy in patients with early cervical cancer. J Obstet Gynaecol Res. 2016 Jan;42(1):77-86. doi: 10.1111/jog.12840.

Response 3:

A new paragraph (number 4) focusing on the effect of CO2 has been added with reference to the publications suggested by the reviewers.

Comments 4:

The effect of surgical volume/ surgical experience. It was a controversial aspect of the LACC trial. Please, discuss the role of surgical experience on the oncological outcomes  in LACC/ in MIS.

Yang Y, Huang Y, Li Z. The Surgeon's Proficiency Affected Survival Outcomes of Minimally Invasive Surgery for Early-Stage Cervical Cancer: A Retrospective Study of 851 Patients. Front Oncol. 2021 Nov 16;11:787198. doi: 10.3389/fonc.2021.787198.

Response 4:

A new paragraph (number 7) focusing on the effect of surgical expertise and learning curve on oncologic outcome has been added.

Comments 5:

One minor remark. The format of the reference list is other than the format of the mdpi-journals. Moreover, the authors use very different formats for references within the same Reference list. Look at e.g. Ref.13,4,15,16 etc. Any of them applies another citation format. It looks a little bit chaotic.  Please, use a template or a reference manager and re-format the reference list accordingly.

Response 5:

The references have been adjusted according to the format of the journal

We hope that the changes made to the manuscript will be to the satisfaction of the Editor and reviewers and that our revised manuscript will be accepted for publication.

With kind regards,

Dr Omar Touhami

Gynecologic Oncology Division, Department of Obstetrics and Gynecology, Centre intégré universitaire de santé et services sociaux CIUSSS du Saguenay—Lac-Saint-Jean, Sherbrooke University, Québec, Canada

Round 2

Reviewer 3 Report

Well done. I acknowledge the new paragraphs on CO2 and surgeon volume/learning curve. I recommend accepting the manuscript as it is. Congratulations!